# Source Knowledge Anchored Regularization for Unsupervised Domain Adaptation

## Abstract

Deep neural networks trained on labeled source-domain samples often experience significant performance drops when used on target domains with different data distributions. Some unsupervised domain adaptation methods (UDA) address this by explicitly aligning the source and target feature distributions; however, enforcing full alignment without target labels can misalign class semantics. We propose Source Knowledge Anchored Regularization (SKAR) for UDA. This unified end-to-end framework transfers discriminative source knowledge via a composite loss on the network outputs, without explicitly enforcing distributional alignment. Our loss comprises of: (1) an adaptation-loss minimizing the entropy on target predictions to boost model confidence by leveraging source domain knowledge; (2) a regularization-loss for penalizing the model when its predictions falls under a few classes, thereby preventing class collapse; (3) a self-supervised-loss enforcing agreement between two strong augmentations of each target sample; and (4) a fidelity loss for anchors learning the source labels while mitigating overfitting. A curriculum learning schedule is applied to gradually shift the optimization focus from source fidelity to target-oriented objectives. Our main contribution is to couple the adaptation and regularization terms; we demonstrate theoretically (via gradient analysis) and empirically (via ablation and hyperparameter studies, and t-SNE visualizations) that these terms interact synergistically. On the Office-Home, Office-31, and VisDA benchmarks, SKAR achieves state-of-the-art performance, while requiring no auxiliary networks.

## 1 Introduction

The performance of deep neural networks often degrades when there is a distribution shift between the training (source) and test (target) domains. Examples in computer vision include but are not limited to differences in illumination condition, sensor and modality, pose and viewpoint, and image resolution and quality Patel et al. (2015). This phenomenon, commonly referred to as domain shift, violates the i.i.d. assumption and poses a significant challenge to practical deployment in real-world settings, where collecting and annotating large-scale labeled data for every new target domain is infeasible, even though substantial labeled data are available from related domains with different input distributions Wang & Deng (2018). Directly training on the target domain is mostly considered infeasible in practice due to the lack of labels and the scarcity of data. Instead, we need to leverage labeled source data alongside unlabeled target data to learn a model that generalizes to the target domain. However, the domain gap between source and target inputs continues to pose a challenge. Unsupervised domain adaptation (UDA) addresses this setting by leveraging labeled source data with unlabeled target data to train models that generalize to the target domain.

Conventional UDA methods often seek domain-invariant representations, inspired by domain generalization literature. Many approaches, e.g., Ganin et al. (2016); Tzeng et al. (2017), align source and target distributions in the feature space; however, shifts in the input distribution are often heterogeneous across classes, so a single global alignment can distort class boundaries and increase inter-class confusion. Intra-class variability in the target domain can increase substantially, while inter-class separation can shrink, yielding overlapping feature supports. Models trained on the source often rely on domain-specific or spurious correlations that are unstable across domains, so cues that were predictive in the source may be non-predictive or misleading in the target. To address these issues, some approaches, such as Pei et al. (2018), perform class-aware alignment using esti-

mated target labels (pseudo-labels). However, training with pseudo-labels without regularization is prone to a degenerate local minimum, which we call class collapse, where the model assigns most target samples to a small subset of classes and the class-conditional distributions become misaligned.

To avoid class collapse and more effectively leverage source knowledge, we propose an augmented loss that harnesses the model's predictive distribution on unlabeled target data and regularizes it with prior knowledge that every class appears in the target domain. In this paper, we introduce two key components of our augmented loss: Adaptation Loss and Regularization Loss. Adaptation Loss reduces the entropy of the predictive distribution on unlabeled target data, sharpening the model's target predictions. Regularization Loss counters collapse by encouraging high entropy over class-mean probabilities, penalizing the concentration of target predictions into a small subset of classes, and preserving class diversity. We analyze the two proposed loss terms, both mathematically and experimentally, to demonstrate their synergistic behavior; we then tune hyperparameters to balance the loss terms and conduct an ablation study to quantify each component's contribution. We evaluate the full setup on standard unsupervised domain adaptation benchmarks. SKAR achieves an average accuracy of 72% on Office-Home, matching the state of the art, and 90% on Office-31 and 82.2% on VisDA, both surpassing existing methods.

## 2 RELATED WORKS

Many UDA methods focus on domain alignment. Ganin et al. (2016) used a domain discriminator and an adversarial setting to make domains indistinguishable. Pei et al. (2018) used the idea of adversarial trainingGanin et al. (2016) and created a discriminator for each class to accomplish a better domain alignment by reducing class-specific domain discrepancy. Tzeng et al. (2017) propose ADDA, which employs discriminative modeling with untied source and target encoders and a GAN loss to adversarially align target features to a fixed source feature space. Long et al. (2015) introduce the Deep Adaptation Network (DAN), which learns domain-invariant features by embedding latent source and target representations into a reproducing kernel Hilbert space (RKHS) and aligning their mean embeddings via a multi-kernel maximum mean discrepancy (MMD) criterion Gretton et al. (2012). DAN then optimizes the kernel weights through quadratic programming and integrates a linear-time, unbiased MMD estimator into stochastic gradient descent (SGD). Sun et al. (2016) propose CORAL, aligning source and target distributions by computing their covariance matrices and applying a closed-form whitening and re-coloring transformation to match second-order statistics; Sun & Saenko (2016) adapt this to deep networks with Deep CORAL, which incorporates a differentiable loss on feature covariances to encourage domain-invariant representations. Kang et al. (2019) introduce Contrastive Domain Discrepancy (CDD), a class-aware MMD extension that minimizes intra-class and maximizes inter-class divergence across domains. Their Contrastive Adaptation Network (CAN) framework alternates between clustering unlabeled target features for pseudo-labels and minimizing CDD via class-aware sampling. Li et al. (2018) assumed that the batch normalization layers Ioffe & Szegedy (2015) could capture the distribution shift between domains; consequently, they mitigated the gap between domains by utilizing different batch normalization layers for each domain. Also, Maria Carlucci et al. (2017) used a similar idea and proposed domain alignment layers for UDA. Tang et al. (2020) used the idea of balanced pseudo-label assignment inspired by Dizaji et al. (2017) to avoid assigning most of the data into a few classes, and other classes have fewer data points. Also, they use the idea of learnable class centers, which are learned simultaneously through backpropagation. Zhang et al. (2023) propose an upper bound for the target error that incorporates joint error. They minimize this upper bound to learn domain-invariant features while addressing joint distribution mismatches. Also, Zou et al. (2019) introduce a confidence-regularized approach that relies on pseudo-labels in the loss.

## 3 METHODOLOGY

Suppose the source domain dataset is denoted as $\mathcal{D}^s = \{(x_i^s, y_i^s)\}_{i=1}^N$, comprising $N$ sample-label pairs, where $x_i^s$ represents an input image from the source domain and $y_i^s$ its corresponding ground-truth label. Conversely, the target domain dataset is defined as $\mathcal{D}^t = \{x_j^t\}_{j=1}^M$, consisting of $M$ unlabeled samples, where $x_j^t$ denotes an input image from the target domain. Let $f : \mathcal{X} \to \mathcal{P}$ be the full model with a final softmax, where $\mathcal{P} = \left\{ p \in [0,1]^K \mid \sum_{k=1}^K p_k = 1 \right\}$. For any input $x$, $f(x)$

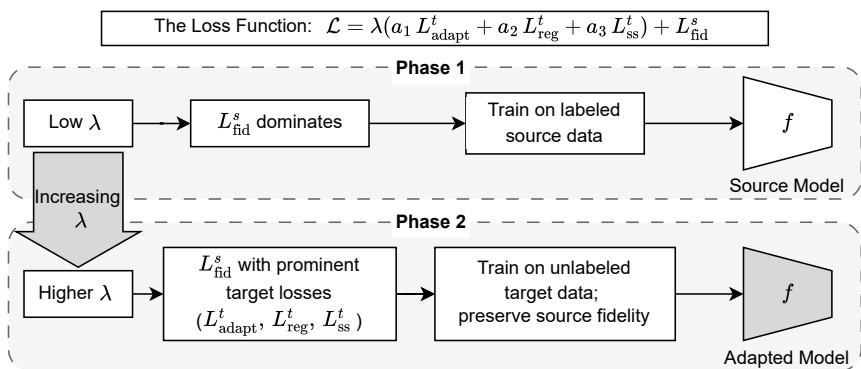

Figure 1: Training overview. In early epochs, $\lambda$ is low, so $L_{\text{fid}}^s$ (Eq. 5) dominates while target-oriented terms are present but down-weighted. As $\lambda$ increases, target losses become prominent and contribute significantly: $L_{\text{adapt}}^t$ (Eq. 2), $L_{\text{reg}}^t$ (Eq. 3), and $L_{\text{ss}}^t$ (Eq. 4) receive larger weights ($\lambda_i = a_i \lambda$, $i \in \{1, 2, 3\}$), guiding adaptation while preserving source fidelity.

returns a $K$-dimensional probability vector, where $K$ is the number of classes. For target and source samples we set $p_i^t = f(x_i^t)$, $p_{i,k}^t = [f(x_i^t)]_k$ and $p_i^s = f(x_i^s)$, $p_{i,k}^s = [f(x_i^s)]_k$, respectively. Here, $p_i^t$ and $p_i^s$ are the $K$-dimensional probability vectors for the $i$-th target and source images, respectively, and $p_{i,k}^t, p_{i,k}^s$ are their $k$-th components, representing the predicted probability that the $i$-th target image and the $i$-th source image in, respectively, belong to class $k$. Consequently, $P^t = [\, p_1^t \,;\, \ldots \,;\, p_{n^t}^t \,] \in \mathbb{R}^{n^t \times K}$, where $n^t$ is the number of target samples in the current mini-batch; thus, $P^t$ stacks the batch's probability vectors (one per row).

### 3.1 SOURCE KNOWLEDGE ANCHORED REGULARIZATION LOSS

Our method employs a four-term loss function, where each term addresses a specific objective; collectively, these terms promote the model to adapt its knowledge from source to target. The approach operates in an end-to-end manner, relying solely on the model's outputs, thereby facilitating integration with diverse architectures. The overall training objective is defined as

$$\mathcal{L} = \lambda_1 L_{\text{adapt}}^t + \lambda_2 L_{\text{reg}}^t + \lambda_3 L_{\text{ss}}^t + L_{\text{fid}}^s, \quad \lambda_i = a_i \, \lambda, \quad i \in \{1, 2, 3\}. \tag{1}$$

Substituting $\lambda_i = a_i \, \lambda$ into 1 and factoring out $\lambda$ yields $\mathcal{L} = \lambda(a_1 \, L_{\text{adapt}}^t + a_2 \, L_{\text{reg}}^t + a_3 \, L_{\text{ss}}^t) + L_{\text{fid}}^s$. A detailed description of the hyperparameter search for $\{a_i\}$ is given in Section 3.3 and Appendix B. In all reported experiments, we fix $a_1 = a_2 = a_3 = 0.5$. An overview of the training flow and loss weighting is shown in Fig. 1. We initialize $\lambda$ at 0 and increase it during training according to the schedule described in Section 4.2. This biases training in the early epochs toward the source-domain fidelity term, with $L_{\text{fid}}^s$ dominating, so the network first learns from labeled source data. As $\lambda$ grows, more weight is placed on the target-oriented terms in Eq. 1. The entropy-minimization term $L_{\text{adapt}}^t$ encourages confident predictions on target samples, while the regularization term $L_{\text{reg}}^t$ suppresses degenerate solutions where the model assigns the majority of target instances to only a few classes.

**Adaptation Loss.** Since target-domain labels are unavailable, we exploit knowledge learned from the source domain. During the initial epochs the model is trained primarily on labeled source data (as discussed above). In later epochs the model makes predictions on target data based on the source-domain knowledge. We then adapt by minimizing the entropy of those predictions. Specifically, we add the entropy of the target predictive distribution to the loss:

$$L_{\text{adapt}}^t = \mathrm{H}(P^t) = -\frac{1}{n^t} \sum_{i=1}^{n^t} \sum_{k=1}^{K} p_{i,k}^t \, \log p_{i,k}^t \tag{2}$$

Minimizing $L_{\text{adapt}}^t$ increases the confidence of the model's predictions on target samples.

**Regularization Loss.** Applying a source-trained model to target data can collapse predictions onto a few classes, leaving others sparsely populated. To counter this, we regularize model predictions.

$$L_{\text{reg}}^t = -\text{H}(g^t) = -\sum_{k=1}^{K} g_k^t \log g_k^t, \quad g^t = (g_1^t, \ldots, g_K^t)^\top, \quad g_k^t = \frac{1}{n^t}\sum_{i=1}^{n^t} p_{i,k}^t. \quad (3)$$

where $g^t$ is the mean class-probability vector over a target mini-batch of $K$.

In the collapsed scenario, a few classes have larger $g_k^t$ values, while others have smaller ones. Even in this collapsed regime, the outputs still carry source information. For example, the top-1 prediction may be incorrect due to the collapse, yet the correct label can appear among the next most probable classes. To tackle this problem, we introduce a term based on the model predictions to penalize the output mostly when the model predicts the majority of batch data points as a few classes. In other words, this loss term is informed by our prior knowledge that the target labels are not confined to a small subset of classes. Minimizing $L_{\text{reg}}^t$, thus maximizes the entropy of $g^t$, encourages the model to distribute assignments among classes and to spread probability mass over multiple top-ranked hypotheses, as demonstrated by the gradient analysis in section 3.2. This, in turn, steers optimization away from suboptimal local minima associated with class collapse (i.e., predictions concentrated in a few classes) and enables the model to exploit the richer supervisory signal encoded in the full ranking of class scores for each data point. Our experiments across multiple UDA benchmarks, including imbalanced settings, show that the regularization term is robust to class imbalance.

**Self-Supervised Loss.** This term is a self-supervised loss that ensures the model learns a robust representation, making it invariant to image noise Chen et al. (2020); Caron et al. (2021).

$$L_{\text{ss}}^t = \text{KL}(P^t \,\|\, P^{t,Aug2}) = \frac{1}{n^t}\sum_{i=1}^{n^t}\text{KL}(p_i^t, p_i^{t,Aug2}) = \frac{1}{n^t}\sum_{i=1}^{n^t}\sum_{k=1}^{K} p_{i,k}^t \log\left(\frac{p_{i,k}^t}{p_{i,k}^{t,Aug2}}\right) \quad (4)$$

where, $P^{t,Aug2}$ is the stack of probability vectors obtained from the same batch of target images as $P^t$ but processed with the second augmentation, whereas $P^t$ uses the base augmentation; their discrepancy is measured by the Kullback-Leibler (KL) divergence.

By minimizing the KL divergence between the model's outputs on each base-augmented image and its grayscale-augmented counterpart, we encourage the network to leverage both sketch-like and color-based cues. Sketch-like patterns offer simplicity and greater generalization, while color-based patterns carry richer information and are more like the source data. This dual focus combines the robustness of structural features with the discriminative strength of color variations, leading to more accurate predictions. For the digit datasets, which are already grayscale and sensitive to horizontal flipping, we employ a different augmentation strategy described in the Appendix C.

**Fidelity Loss.** This loss is cross-entropy with label smoothing and relates to the source domain.

$$L_{\text{fid}}^s = -\frac{1}{n^s}\sum_{i=1}^{n^s}\sum_{k=1}^{K}\left[(1-\epsilon)\cdot\mathbf{I}[k=y_i^s]+\frac{\epsilon}{K}\right]\log p_{i,k}^s \quad (5)$$

where $\mathbf{I}$ is the indicator function. Experiments have shown that cross-entropy with label smoothing can improve generalization and prevent overfitting Szegedy et al. (2016); Müller et al. (2019). We use $\epsilon = 0.2$ to improve domain generalization, consequently improving UDA accuracy. In the early epochs of the training, the model learn the labeled source domain using this term and does not overfit to the source domain. During other epochs, this term maintains high source-domain accuracy and ensures that the entire training process converges to a local minimum faithful to the labeled source knowledge; this matters because only the source domain has ground-truth labels.

### 3.2 Gradients Analysis

To analyze the effect of $L_{\text{adapt}}^t$ and $L_{\text{reg}}^t$ and show their synergistic behavior, we compute the partial derivatives of each with respect to the logits $z_{i,k}^t$ and examine their individual and combined effects:

$$\frac{\partial L_{\text{adapt}}^t}{\partial z_{i,k}^t} = -\frac{p_{i,k}^t}{N}\left[\log p_{i,k}^t + \text{H}(p_i^t)\right] \quad (6) \qquad\qquad \frac{\partial L_{\text{reg}}^t}{\partial z_{i,k}^t} = \frac{p_{i,k}^t}{N}\left[\log g_k^t + \text{H}(p_i^t, g^t)\right] \quad (7)$$

Gradient of $L_{\text{adapt}}^t + L_{\text{reg}}^t$ w.r.t. logits $z_{i,k}^t$:
$$\frac{\partial(L_{\text{adapt}}^t + L_{\text{reg}}^t)}{\partial z_{i,k}^t} = \frac{p_{i,k}^t}{N}\left[\log\frac{g_k^t}{p_{i,k}^t} + \text{KL}(p_i^t\,\|\,g^t)\right] \quad (8)$$

Detailed calculations of the derivatives of Eqs. 6, 7 and 8 are provided in Appendix A. In Fig. 2 we plot the gradient of $L_{\text{adapt}}^t + L_{\text{reg}}^t$ w.r.t. logits $z_{i,k}^t$ (Eq. 6), as a function of the predicted probability of class 1 for the first data point in the batch ($p_{1,1}^t$) in a five-class task. The remaining probability mass $(1 - p_{1,1}^t)$ is equally shared among the other four classes of the first data point. Each panel uses a different mean-class-probability vector $g^t$: **(a) Uniform**: $g_k^t = 0.2$ for all $k$; **(b) Class-1 dominant**: $g_1^t = 0.6,\ g_{k\neq1}^t = 0.1$; **(c) Class-1 submissive**: $g_1^t = 0.1,\ g_{k\neq1}^t = 0.225$.

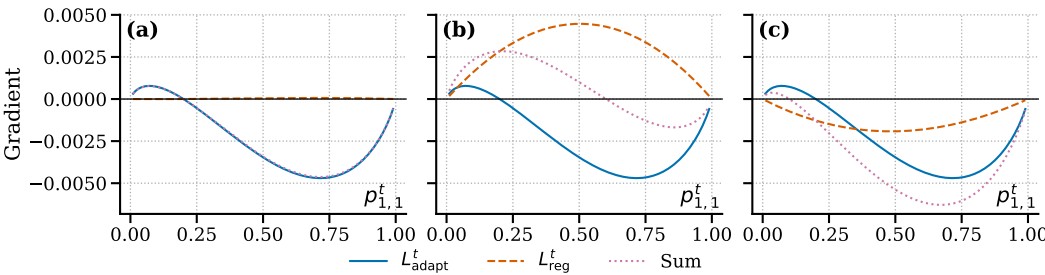

Figure 2: Gradients of $L_{\text{adapt}}^t$, $L_{\text{reg}}^t$, and their sum with respect to the logits $z_{i,k}^t$ are shown on the y-axis, plotted against the predicted probability of class 1 for the first data point in the batch, $p_{1,1}^t$, on the x-axis. The plots correspond to a five-class task. Results are shown for three mean-class-probability vectors $g^t$: (a) uniform, (b) class-1 dominant, and (c) class-1 submissive.

In all three scenarios, the gradient of the adaptation loss is identical (since it does not depend on $g^t$); the differences between panels arise solely from the regularization gradient.

**Fig. 2(a)** corresponds to the uniform scenario. Here $p_{1,1}^t = 0.2$ exactly satisfies the regularization term, and when $p_{1,1}^t \neq 0.2$ the regularization term's effect is negligible (because probability masses of other data points are distributed evenly among all of the classes). Consequently, the total gradient reduces to the adaptation term alone: for $p_{1,1}^t < 0.2$ the total gradient is positive (decreasing $z_{1,1}^t$), and for $p_{1,1}^t > 0.2$ it is negative (increasing $z_{1,1}^t$). The same logic applies to each $z_{1,k}^t$ for $k \neq 1$.

**Fig. 2(b)** illustrates the class-1 dominant case ($g_1^t = 0.6,\ g_{k\neq1}^t = 0.1$). The zero of the summation of the gradient shifts to $p_{1,1}^t = 0.6$, raising the threshold at which the total gradient changes sign. As a result, only data points with $p_{1,1}^t > 0.6$ receive a negative total gradient (increasing $z_{1,1}^t$), whereas those with $p_{1,1}^t < 0.6$ receive a positive total gradient (decreasing $z_{1,1}^t$), making it harder for class 1 to accumulate assignments that are not high-confidence during training.

**Fig. 2(c)** shows the class-1 submissive case ($g_1^t = 0.1,\ g_{k\neq1}^t = 0.225$). Here the zero-crossing threshold moves down to $p_{1,1}^t = 0.1$, so the total gradient is positive for $p_{1,1}^t < 0.1$ (decreasing $z_{1,1}^t$) and negative for $p_{1,1}^t > 0.1$ (increasing $z_{1,1}^t$). This lower threshold makes it easier for the submissive class to gain assignments during training. It means even moderately confident assignments ($p_{1,1}^t > 0.1$) are reinforced, provided they are not strongly predicted for other classes.

### 3.3 HYPERPARAMETER SEARCH

In Fig. 3, we present the results of a search over the weighting coefficients $a_1$ and $a_2$ on the Office-31 benchmark. Each point corresponds to a complete $(a_1, a_2)$ configuration evaluated across the six source→target tasks, with the color reflecting the mean target classification accuracy achieved across all tasks. As shown in Fig. 3(b), the best performance across epochs occurs primarily along

the line where $a_1 = a_2$, highlighting the importance of weighting the terms $L_{\mathrm{adapt}}^t$ and $L_{\mathrm{reg}}^t$ equally. Additionally, in Fig. 3(a), we observe that the highest final accuracy is achieved for lower values of both $a_1$ and $a_2$, again near the $a_1 = a_2$ line. This indicates that for higher values of $a_1$ and $a_2$, training tends to diverge in the middle stages, despite achieving high accuracy at that point.

This divergence attributes to the interplay between the terms $L_{\mathrm{adapt}}^t$, $L_{\mathrm{reg}}^t$, and $L_{\mathrm{fid}}^s$. During the early stages of training, the influence of $L_{\mathrm{fid}}^s$ (the only supervised term) is significant due to its high weighting. However, as training progresses and the $\lambda$ factor increases, if $a_1$ and $a_2$ are set too high, the combined energies of $L_{\mathrm{adapt}}^t$ and $L_{\mathrm{reg}}^t$ begin to dominate, diminishing the role of $L_{\mathrm{fid}}^s$. Since $L_{\mathrm{fid}}^s$ is crucial for preserving correct features, its diminished influence causes instability, leading to divergence in the training process. Additional plots and full results of the search over $a_1$ and $a_2$ are provided in the Appendix B.

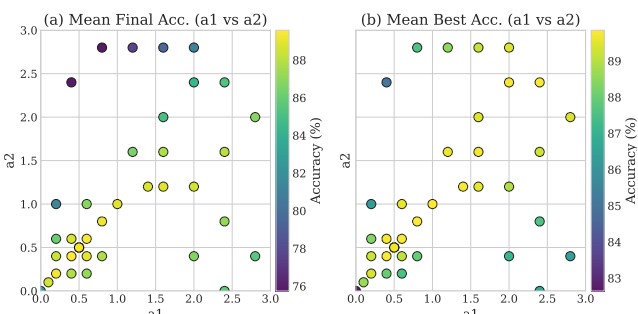

Figure 3: Search over the weighting coefficients $a_1$ and $a_2$ on the OFFICE-31 benchmark. Each dot represents a complete $(a_1, a_2)$ configuration across the six source→target tasks, with the color indicating its mean target classification accuracy across all tasks. (a) Mean *final* accuracy measured at the last training epoch. (b) Mean *best* accuracy, i.e. the highest target-domain accuracy achieved at any epoch.

The self-supervised loss term can be written as $L_{\mathrm{ss}}^t = \mathrm{KL}\big(P^t \parallel P^{t,Aug2}\big) = \mathrm{H}\big(P^t, P^{t,Aug2}\big) - \mathrm{H}\big(P^t\big)$, where $\mathrm{H}(p,q) = -\sum_i p_i \log q_i$ denotes the cross-entropy between two distributions. Consequently, $L_{\mathrm{ss}}^t$ has an energy scale comparable to $L_{\mathrm{adapt}}^t$ and $L_{\mathrm{reg}}^t$. For consistency across objectives, we therefore fix $a_1 = a_2 = a_3 = 0.5$ in all subsequent experiments.

## 3.4 SYNERGY OF ADAPTATION AND REGULARIZATION IN CLASS ALLOCATION

To investigate the impact of combining the adaptation and regularization terms, we conducted experiments to show how they affect the improvement of data point-to-class allocations.

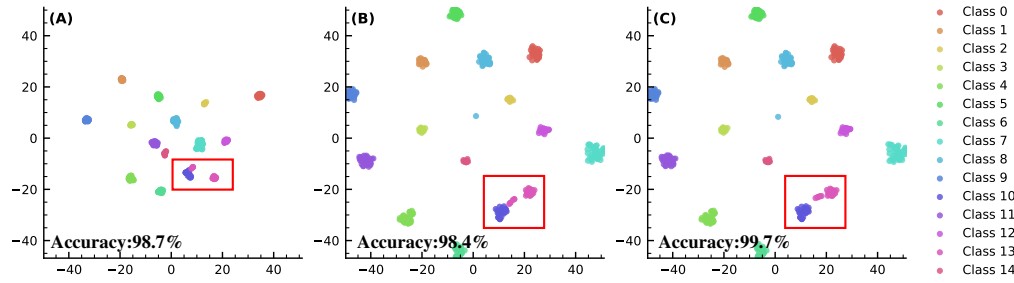

Figure 4: t-SNE visualizations of target domain (Webcam) data points, using DSLR as the source domain with 15 classes from Office-31, under three training scenarios: (A) source-only model trained supervisedly on the source domain (DSLR); (B) source-only model further trained with unlabeled target data using only the adaptation loss term; (C) source-only model further trained with both adaptation and regularization loss terms. The red rectangles highlight clusters where adaptation alone fails to separate mixed classes, whereas the combination of adaptation and regularization resolves the overlap, leading to improved classification.

In Fig. 4, we illustrate three scenarios. First, we trained a source-only model supervisedly on the source domain (DSLR), then applied it to the target domain (Webcam) data points and display the t-SNE projection of the outputs in Fig. 4(A). Second, we further trained the source-only model with unlabeled target data points using only the adaptation loss term and display the t-SNE projection of

| METHOD | A→C | A→P | A→R | C→A | C→P | C→R | P→A | P→C | P→R | R→A | R→C | R→P | Avg |
|---|---|---|---|---|---|---|---|---|---|---|---|---|---|
| Source Only | 34.9 | 50.0 | 58.0 | 37.4 | 41.9 | 46.2 | 38.5 | 31.2 | 60.4 | 53.9 | 41.2 | 59.9 | 46.1 |
| DANN (2016) | 45.6 | 59.3 | 70.1 | 47.0 | 58.5 | 60.9 | 46.1 | 43.7 | 68.5 | 63.2 | 51.8 | 76.8 | 57.6 |
| MCD (2018) | 51.9 | 70.7 | 74.8 | 59.0 | 68.4 | 68.8 | 58.2 | 51.6 | 75.1 | 69.5 | 55.8 | 79.3 | 65.3 |
| CDAN (2018) | 50.7 | 70.6 | 76.0 | 57.6 | 70.0 | 70.0 | 57.4 | 50.9 | 77.3 | 70.9 | 56.7 | 81.6 | 65.8 |
| ADA (2019) | 50.1 | 63.4 | 70.9 | 56.6 | 66.5 | 65.9 | 54.7 | 51.5 | 74.2 | 66.8 | 54.9 | 77.6 | 62.8 |
| SymNets (2019) | 47.7 | 72.9 | 78.5 | 64.2 | 71.3 | 74.2 | 64.2 | 48.8 | 79.5 | 74.5 | 52.6 | 82.7 | 67.6 |
| SPL (2020) | 54.5 | 77.8 | 81.9 | 65.1 | 78.0 | 81.1 | 66.0 | 53.1 | 82.8 | 69.9 | 55.3 | 86.0 | 71.0 |
| AADA (2020) | 54.0 | 71.3 | 77.5 | 60.8 | 70.8 | 71.2 | 59.1 | 51.8 | 76.9 | 71.0 | 57.4 | 81.8 | 67.0 |
| SCAL (2022) | 55.3 | 72.7 | 78.7 | 63.1 | 71.7 | 73.5 | 61.4 | 51.6 | 79.9 | 72.5 | 57.8 | 81.0 | 68.3 |
| MJE (2023) | 60.3 | 77.8 | 81.0 | 66.0 | 74.4 | 74.5 | 66.7 | 59.3 | 81.8 | 74.2 | 62.7 | 84.9 | 72.0 |
| TAROT (2025) | 55.7 | 73.9 | 77.8 | 60.1 | 73.1 | 69.9 | 59.6 | 55.0 | 78.7 | 71.9 | 59.5 | 84.4 | 68.3 |
| **OURS** | 57.1 | 78.0 | 81.7 | 67.2 | 76.8 | 78.9 | 66.7 | 56.1 | 82.4 | 74.6 | 59.6 | 85.2 | 72.0 |

Table 1: Accuracy comparison (%) of domain adaptation methods on the Office-Home dataset over 12 adaptation tasks across four domains: A (Art), C (Clipart), P (Product), and R (Real World). The best result in each column is bold; the second best is underlined.

the target outputs in Fig. 4(B). Finally, we trained the source-only model with unlabeled target data points using both the adaptation and regularization loss terms and display the t-SNE projection of the target outputs in Fig. 4(C).

The red rectangle in Fig. 4(A) shows two clusters of data points, where one cluster consists of two classes. This overlap lowers classification accuracy. In Fig. 4(B), where the source-only model is trained with the adaptation loss term only, the red rectangle highlights the same clusters with the same issue. However, in Fig. 4(C), where the source-only model is trained with both adaptation and regularization terms, the red rectangle shows the same clusters, now correctly separated as the mixed data points have moved to their proper cluster. These scenarios demonstrate that the combination of adaptation and regularization terms works synergistically, leading to improved classification. This synergy arises because the regularization term penalizes the dominant class, allowing other class probabilities that are already high to increase further and correct the model's predictions. Thus, the combination of the two terms leverages the full predictive distribution, exploiting more information from the model's outputs to improve classification.

## 4 EXPERIMENTS

We evaluate our method on four UDA benchmarks: Office-31 Saenko et al. (2010) and Office-Home Venkateswara et al. (2017), the VisDA-2017 Peng et al. (2017), and three digit-recognition datasets: MNIST (Lecun et al., 1998), USPS (Hull, 2002), and SVHN (Netzer et al., 2011).

These benchmarks span a spectrum of realistic domain shifts, including changes in capture conditions, variations in style and background, and substantial stylistic discrepancies in appearance; this diversity makes them representative and widely adopted testbeds in the UDA literature. Together, they present varied scenarios for robustly evaluating

| METHOD | plane | bcycl | bus | car | horse | knife | mcycl | person | plant | sktbrd | train | truck | Avg |
|---|---|---|---|---|---|---|---|---|---|---|---|---|---|
| Source Only | 55.1 | 53.3 | 61.9 | 59.1 | 80.6 | 17.9 | 79.7 | 31.2 | 81.0 | 26.5 | 73.5 | 8.5 | 52.4 |
| MDD (2015) | 87.1 | 63.0 | 76.5 | 42.0 | 90.3 | 42.9 | 85.9 | 53.1 | 49.7 | 36.3 | 85.8 | 20.7 | 61.1 |
| DANN (2016) | 81.9 | 77.7 | 62.8 | 44.3 | 81.2 | 29.5 | 65.1 | 28.6 | 51.9 | 54.5 | 82.8 | 7.8 | 57.4 |
| MCD (2018) | 87.0 | 60.9 | 83.7 | 64.0 | 88.9 | 79.6 | 84.7 | 76.9 | 68.8 | 40.3 | 83.0 | 25.8 | 71.9 |
| GPDA (2019) | 83.0 | 74.3 | 84.0 | 66.0 | 87.6 | 78.3 | 88.3 | 73.1 | 90.1 | 57.3 | 80.2 | 39.7 | 74.5 |
| CRST (2019) | 88.0 | 79.2 | 61.0 | 60.0 | 87.5 | 81.4 | 86.3 | 78.8 | 85.6 | 86.6 | 73.9 | 68.8 | 78.1 |
| MCC (2020) | 88.1 | 80.3 | 80.5 | 71.5 | 90.4 | 93.2 | 85.0 | 71.6 | 89.4 | 73.8 | 85.0 | 36.9 | 78.8 |
| MJE (2023) | 93.8 | 79.5 | 79.3 | 55.9 | 93.9 | 93.8 | 86.5 | 80.3 | 91.6 | 87.7 | 85.4 | 51.6 | 81.6 |
| TAROT (2025) | - | - | - | - | - | - | - | - | - | - | - | - | 67.1 |
| **OURS** | 93.3 | 75.9 | 83.8 | 70.1 | 93.1 | 88.8 | 92.7 | 76.7 | 90.2 | 83.3 | 89.2 | 49.1 | **82.2** |

Table 2: Accuracy per class (%) comparison on the VisDA Dataset (synthetic→Real). Bold for the best and underline for the second-best.

the contribution of our proposed loss terms and for fair comparison with state-of-the-art methods. We provide the mean and standard deviation of accuracy in all cases, with the exception of Office-Home and VisDA, for which only the mean is reported to align with previous studies.

We compare our model with several domain adaptation methods. Performance comparisons on the Office-Home, VisDA, Office-31, and Digits (SVHN, MNIST, USPS) datasets are shown in Tables 1,

2, 3 and 4, respectively. Some UDA methods are built upon and depend on other UDA methods (e.g., Na et al. (2021); Gu et al. (2020)); for fair comparisons, we focus on standalone methods.

We follow the standard setting: for Office-31 and Office-Home, we use a ResNet-50 backbone pretrained on ImageNet He et al. (2015); Deng et al. (2009); for VisDA, we use a ResNet-101 backbone pretrained on ImageNet. The model architecture for the Digits experiments is described in Appendix C.

On **Office-31**, our method achieves the best average accuracy (90.0%), outperforming recent methods. This underscores the strength of our unified framework.

On **Office-Home**, our method achieves the highest average accuracy (72.0%), matching state of the art and demonstrating generalization across diverse source–target pairs.

On **VisDA**, our method achieves the best average accuracy (82.2%), outperforming recent methods. This evidences the strength of our framework.

About **Digits (SVHN, MNIST, USPS)**, these digit benchmarks are saturated, with recent methods typically achieving accuracies between 95% and 99%. Our method delivers high and consistent performance: it attains the best result on *SVHN→MNIST* (99.2±0.1) and the second-best on *USPS→MNIST* (96.7±0.5). For *MNIST→USPS* and *MNIST* *→USPS**, our approach remains competitive (95.5±0.4 and 97.3±0.3) relative to top-performing methods.

Overall, these experimental results clearly demonstrate the effectiveness of our proposed model for addressing challenging domain adaptation tasks, confirming its superiority and generalization. Furthermore, compared to CRST Zou et al. (2019), our approach achieves higher accuracy, highlighting the importance of using the full predictive distribution of data points in regularization.

| METHOD | A→W | D→W | W→D | A→D | D→A | W→A | AVG |
|---|---|---|---|---|---|---|---|
| Source Only | 68.4±0.2 | 96.7±0.1 | 99.3±0.1 | 68.9±0.2 | 62.5±0.3 | 60.7±0.3 | 76.1 |
| DANN (2016) | 82.0±0.4 | 96.9±0.2 | 99.1±0.1 | 79.7±0.4 | 68.2±0.4 | 67.4±0.5 | 82.2 |
| ADDA (2017) | 86.2±0.5 | 96.2±0.3 | 98.4±0.3 | 77.8±0.3 | 69.5±0.4 | 68.9±0.5 | 82.9 |
| MCD (2018) | 88.6±0.2 | 98.5±0.1 | 100.0±0.0 | 92.2±0.2 | 69.5±0.1 | 69.7±0.3 | 86.5 |
| CDAN (2018) | 94.1±0.1 | 98.6±0.1 | 100.0±0.0 | 92.9±0.2 | 71.0±0.3 | 69.3±0.3 | 87.7 |
| SymNets (2019) | 90.8±0.1 | 98.8±0.3 | 100.0±0.0 | 93.9±0.5 | 74.6±0.6 | 72.5±0.5 | 88.4 |
| CRST (2019) | 89.4±0.7 | 98.9±0.4 | 100.0±0.0 | 88.7±0.8 | 72.6±0.7 | 70.9±0.5 | 86.8 |
| SPL (2020) | 92.7±0.0 | 98.1±0.0 | 99.8±0.0 | 93.7±0.0 | 76.4±0.0 | 76.9±0.0 | 89.6 |
| MCC (2020) | **95.5±0.2** | 98.6±0.1 | 100.0±0.0 | **94.4±0.3** | 72.9±0.2 | 74.9±0.3 | 89.4 |
| SCAL (2022) | 93.5±0.2 | 98.5±0.1 | 100.0±0.0 | 93.4±0.3 | 72.4±0.1 | 74.0±0.3 | 88.6 |
| MJE (2023) | 91.9±0.5 | **99.0±0.2** | 100.0±0.0 | 93.7±0.5 | 76.1±0.2 | **77.8±0.2** | 89.8 |
| TAROT (2025) | **95.5** | 98.0 | **100.0** | 94.2 | 74.5 | 73.7 | 89.3 |
| **OURS** | 94.4±0.8 | 98.9±0.2 | 100.0±0.0 | 93.4±0.5 | **77.5±0.4** | 76.0±0.9 | **90.0** |

Table 3: Accuracy comparison (%) of domain adaptation methods on the Office-31 dataset over six transfer tasks across three domains: A (Amazon), D (DSLR), and W (Webcam). The best result in each column is bold; the second best is underlined.

| METHOD | SVHN to MNIST | MNIST to USPS | MNIST* to USPS* | USPS to MNIST |
|---|---|---|---|---|
| Source Only | 67.1 | 76.7 | 79.7 | 63.4 |
| MDD (2013) | 71.1 | – | 81.1 | – |
| DANN (2016) | 71.1 | 77.3 | 85.1 | 73.2 |
| DRCN (2016) | 82.0 ± 0.1 | 91.8 ± 0.1 | – | 73.7 ± 0.1 |
| ADDA (2017) | 76.0 ± 1.8 | 89.4 ± 0.2 | – | 90.1 ± 0.8 |
| MCD (2018) | 96.2 ± 0.4 | 94.2 ± 0.7 | 96.5 ± 0.3 | 94.1 ± 0.3 |
| GPDA (2019) | 98.2 ± 0.1 | 96.5±0.2 | **98.1±0.1** | 96.4 ± 0.1 |
| MJE (2023) | 98.6±0.1 | **96.8±0.2** | 97.9±0.1 | **96.9±0.1** |
| ours | **99.3±0.1** | 95.5 ± 0.4 | 97.8 ± 0.2 | 96.7±0.5 |

Table 4: comparison of domain adaptation methods on the digit datasets over four transfer tasks across four domains: SVHN, MNIST, and USPS. The best result in each column is bold; the second best is underlined.

| Loss Terms | A→W | D→W | W→D | A→D | D→A | W→A | AVG |
|---|---|---|---|---|---|---|---|
| Source Only | 68.4±0.2 | 96.7±0.1 | 99.3±0.1 | 68.9±0.2 | 62.5±0.3 | 60.7±0.3 | 76.1 |
| w/o $L_{\mathrm{reg}}^t$ | 91.2±1.8 | **99.1±0.1** | **100.0±0.0** | 92.0±1.3 | 68.4±0.1 | 64.2±1.6 | 85.8 |
| w/o $L_{\mathrm{adapt}}^t$ | 68.3±1.7 | 96.5±0.3 | 93.2±1.6 | 71.0±1.6 | 57.7±0.7 | 54.8±0.7 | 73.6 |
| w/o $L_{\mathrm{adapt}}^t, L_{\mathrm{reg}}^t$ | 86.9±0.8 | 98.6±0.1 | 99.9±0.1 | 88.8±0.3 | 67.9±0.4 | 65.4±0.4 | 84.6 |
| w/o label smoothing | 91.6±1.0 | 98.2±0.5 | 99.9±0.1 | 89.5±0.8 | 75.7±0.6 | 74.5±0.3 | 88.2 |
| w/o $L_{\mathrm{ss}}^t$ | 93.9±0.4 | 98.7±0.1 | **100.0±0.0** | 92.8±0.3 | 76.4±0.5 | 75.7±0.7 | 89.6 |
| All Terms | **94.4±0.8** | 98.9±0.2 | **100.0±0.0** | **93.4±0.5** | **77.5±0.4** | **76.0±0.9** | **90.0** |

Table 5: Ablation study of loss terms on the Office-31 dataset. The best result in each column is bold; the second best is underlined.

## 4.1 ABLATION STUDY

We conduct an ablation study to analyze the contribution of each loss term to the overall performance using Office-31. The results are in Table 5. Our complete model integrates four losses: adaptation loss ($L_{\text{adapt}}^t$), regularization loss ($L_{\text{reg}}^t$), self-supervised loss ($L_{\text{ss}}^t$), and fidelity loss ($L_{\text{fid}}^s$).

**Effect of Adaptation** ($L_{\text{adapt}}^t$). Omitting $L_{\text{adapt}}^t$ causes the largest accuracy drop ($90.0\% \rightarrow 73.6\%$), confirming that it is indispensable for domain alignment. Without it, the objective does not leverage the model's predictive distribution on unlabeled target data to encourage increased confidence in those predictions. Consequently, no meaningful adaptation occurs. Interestingly, removing both $L_{\text{adapt}}^t$ and $L_{\text{reg}}^t$ simultaneously causes less performance degradation ($90.0\% \rightarrow 84.6\%$), suggesting that $L_{\text{reg}}^t$ alone might excessively regularize the model, degrading its accuracy on the target data.

**Effect of Regularization** ($L_{\text{reg}}^t$). Removing the regularization term $L_{\text{reg}}^t$ leads to a significant performance drop ($90.0\% \rightarrow 85.8\%$). This confirms its role in discouraging prediction collapse into a narrow set of classes and maintaining class diversity. In other words, $L_{\text{reg}}^t$ injects additional knowledge into the optimization process, guiding the model away from degenerate predictions and thereby improving accuracy.

**Effect of Label Smoothing.** Disabling label smoothing ($\epsilon = 0$) reduces overall accuracy from 90.0% to 88.2%. Label smoothing discourages overfitting on the source domain and thus improves the model's generalization to related target domains. Without it, target accuracy drops, so $L_{\text{adapt}}^t$ operates on a less accurate predictive distribution, further degrading adaptation.

**Effect of Self-supervision** ($L_{\text{ss}}^t$). Omitting the self-supervised loss led to a modest drop in overall accuracy ($90.00\% \rightarrow 89.6\%$), underscoring its complementary role. $L_{\text{ss}}^t$ reduces sensitivity to noise, making adaptation more effective. However, it can be omitted to reduce computational cost, at the cost of a minor drop in performance.

## 4.2 IMPLEMENTATION DETAILS

We follow standard protocols established in unsupervised domain adaptation (UDA). For all experiments, we utilize an ImageNet-pretrained feature extractor backbone. We train our model using stochastic gradient descent (SGD) with momentum set to 0.9. We set the initial learning rate to 0.01 for MLP layers and 0.001 for the backbone module. We adopt the annealing strategy proposed by Ganin et al. (2016). Let $T$ denote the total number of training epochs. For experiments with the ResNet-50 model pre-trained on ImageNet, we set $T = 50$ when fine-tuning on Office-31 and similarly for Office-Home; for architectures trained from random initialization on the digit datasets, we use $T = 200$. For each epoch $0 \leq t < T$, we use the following learning-rate schedule and, following the same work, also increase $\lambda(t)$ from 0:

$$\alpha(t) = \alpha_0 \left(1 + \frac{t}{20}\right)^{-\beta}, \alpha_0 = 10^{-3}, \ \beta = 0.75, \qquad \lambda(t) = \frac{2}{1 + \exp\left(-\frac{t}{20}\right)} - 1.$$

In each experiment, the number of training steps per epoch is set based on the larger of the source and target datasets, so that each epoch performs a full pass over that dataset.

## 5 CONCLUSION

We introduce an unsupervised domain-adaptation framework that transfers knowledge from labeled source data to unlabeled target data while exploiting the prior knowledge that every class is represented in the target domain. The method combines a label-smoothed Fidelity loss on the source domain with entropy-based Adaptation, diversity-preserving Regularization, and augmentation-consistent Self-Supervision on the target domain. Extensive experiments on standard UDA benchmarks demonstrate the effectiveness of the proposed approach: SKAR achieves 90.0% on Office-31 and 82.2% on VisDA, surpassing existing methods, and 72.0% on Office-Home, matching the state of the art. Ablation results confirm that each loss term contributes substantially to overall accuracy.

## REPRODUCIBILITY STATEMENT

For reproducibility, we provide details on model architectures, data preprocessing, and dataset descriptions in Appendix C. The code package contains an instruction file for running the experiments, anonymous download links for the datasets, and all necessary code files to reproduce our work.

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

## A   GRADIENT CALCULATIONS

**Calculating derivative of $L^t_{\text{adapt}}$ and $L^t_{\text{reg}}$ with respect to the input logits $z^t_{i,k}$:**

Partial derivative of $L^t_{\text{adapt}}$ (2) respect to the input probabilities $p^t_{i,k}$:

$$\frac{\partial L^t_{\text{adapt}}}{\partial p^t_{i,k}} = -\frac{1}{N}\left(\log p^t_{i,k} + 1\right) \tag{9}$$

Softmax function:

$$p^t_{i,k} = \text{softmax}(z^t_i)_k = \frac{\exp\left(z^t_{i,k}\right)}{\displaystyle\sum_{k'=1}^{K} \exp\left(z^t_{i,k'}\right)}.$$

Partial derivative of softmax w.r.t. logits $z^t_{i,k}$:

$$\frac{\partial p^t_{i,k}}{\partial z^t_{i,k'}} = \begin{cases} p^t_{i,k}\left(1 - p^t_{i,k}\right), & \text{if } k = k' \\ -p^t_{i,k}\,p^t_{i,k'}, & \text{if } k \neq k' \end{cases} \tag{10}$$

Partial derivative of $L^t_{\text{adapt}}$ w.r.t. logits $z^t_{i,k}$:

$$\frac{\partial L^t_{\text{adapt}}}{\partial z^t_{i,k}} = \sum_{k'=1}^{K} \frac{\partial L^t_{\text{adapt}}}{\partial p^t_{i,k'}} \frac{\partial p^t_{i,k'}}{\partial z^t_{i,k}}$$

Substituting equations 9 and 10 gives

$$\frac{\partial L^t_{\text{adapt}}}{\partial z^t_{i,k}} = -\frac{1}{N}\left(\log p^t_{i,k} + 1\right) p^t_{i,k}\left(1 - p^t_{i,k}\right)$$

$$+ \sum_{k' \neq k} \frac{1}{N}\left(\log p^t_{i,k'} + 1\right) p^t_{i,k}\, p^t_{i,k'}$$

Factoring out common terms:

$$\frac{\partial L^t_{\text{adapt}}}{\partial z^t_{i,k}} = -\frac{p^t_{i,k}}{N}\left[\left(\log p^t_{i,k} + 1\right)\left(1 - p^t_{i,k}\right)\right.$$

$$\left. - \sum_{k' \neq k}\left(\log p^t_{i,k'} + 1\right) p^t_{i,k'}\right]$$

$$= -\frac{p^t_{i,k}}{N}\left[\left(\log p^t_{i,k} + 1\right) - \left(\log p^t_{i,k} + 1\right)p^t_{i,k}\right.$$

$$\left. - \sum_{k' \neq k}\left(\log p^t_{i,k'} + 1\right)p^t_{i,k'}\right]$$

$$= -\frac{p_{i,k}^t}{N} \left[ (\log p_{i,k}^t + 1) - \sum_{k'} (\log p_{i,k'}^t + 1) p_{i,k'}^t \right]$$

$$= -\frac{p_{i,k}^t}{N} \left[ (\log p_{i,k}^t + 1) - \sum_{k'} p_{i,k'}^t \log p_{i,k'}^t - \sum_{k'} p_{i,k'}^t \right]$$

Noting that

$$-\sum_{k'} p_{i,k'}^t \log p_{i,k'}^t = \text{Entropy}(p_i^t),$$

$$\sum_{k'} p_{i,k'}^t = 1$$

the derivative can be written compactly as

$$\frac{\partial L_{\text{adapt}}^t}{\partial z_{i,k}^t} = -\frac{p_{i,k}^t}{N} \left[ \log p_{i,k}^t + \text{Entropy}(p_i^t) \right]$$

Partial derivative of $L_{\text{reg}}^t$ (3) w.r.t. class mean probabilities $g_k^t$:

$$\frac{\partial L_{\text{reg}}^t}{\partial g_k^t} = \log g_k^t + 1$$

Partial derivative of class mean probability $g_k^t$ w.r.t. per-sample probabilities $p_{i,k}^t$:

$$\frac{\partial g_k^t}{\partial p_{i,k'}^t} = \begin{cases} \frac{1}{N}, & \text{if } k = k' \\ 0, & \text{if } k \neq k' \end{cases}$$

Partial derivative of $L_{\text{reg}}^t$ w.r.t. $p_{i,k}^t$:

$$\frac{\partial L_{\text{reg}}^t}{\partial p_{i,k}^t} = \frac{1}{N} (\log g_k^t + 1)$$

Partial derivative of $L_{\text{reg}}^t$ w.r.t. $z_{i,k}^t$:

$$\frac{\partial L_{\text{reg}}^t}{\partial z_{i,k}^t} = \frac{1}{N} \sum_{k'} (\log g_k^t + 1) \frac{\partial p_{i,k'}^t}{\partial z_{i,k}^t}$$

Substituting equation 10 gives:

$$\frac{\partial L_{\text{reg}}^t}{\partial z_{i,k}^t} = \frac{1}{N} (\log g_k^t + 1) \, p_{i,k}^t \, (1 - p_{i,k}^t)$$

$$+ \sum_{k' \neq k} \frac{1}{N} (\log g_{k'}^t + 1) \, (-p_{i,k}^t \, p_{i,k'}^t)$$

Factoring out common terms:

$$\frac{\partial L_{\text{reg}}^t}{\partial z_{i,k}^t} = \frac{p_{i,k}^t}{N} \left[ (\log g_k^t + 1) \, (1 - p_{i,k}^t) \right.$$

$$\left. - \sum_{k' \neq k} (\log g_{k'}^t + 1) \, p_{i,k'}^t \right]$$

$$= \frac{p_{i,k}^t}{N} \left[ (\log g_k^t + 1) - (\log g_k^t + 1)p_{i,k}^t \right.$$

$$\left. - \sum_{k' \neq k} (\log g_{k'}^t + 1)p_{i,k'}^t \right]$$

$$= \frac{p_{i,k}^t}{N} \left[ (\log g_k^t + 1) - \sum_{k'} (\log g_{k'}^t + 1)p_{i,k'}^t \right]$$

$$= \frac{p_{i,k}^t}{N} \left[ (\log g_k^t + 1) - \sum_{k'} p_{i,k'}^t \log g_{k'}^t - \sum_{k'} p_{i,k'}^t \right]$$

Noting that

$$- \sum_{k'} p_{i,k'}^t \log g_{k'}^t = \text{CrossEntropy}(p_i^t, g^t),$$

$$\sum_{k'} p_{i,k'}^t = 1$$

the derivative can be written compactly as

$$\frac{\partial L_{\text{reg}}^t}{\partial z_{i,k}^t} = \frac{p_{i,k}^t}{N} \left[ \log g_k^t + \text{CrossEntropy}(p_i^t, g^t) \right]$$

**Gradient of $L_{\text{adapt}}^t + L_{\text{reg}}^t$ w.r.t. logits $z_{i,k}^t$**

$$\frac{\partial (L_{\text{adapt}}^t + L_{\text{reg}}^t)}{\partial z_{i,k}^t} = \frac{\partial L_{\text{adapt}}^t}{\partial z_{i,k}^t} + \frac{\partial L_{\text{reg}}^t}{\partial z_{i,k}^t}$$

Using equations 6 and 7 gives:

$$\frac{\partial (L_{\text{adapt}}^t + L_{\text{reg}}^t)}{\partial z_{i,k}^t} = \frac{p_{i,k}^t}{N} \left[ \log g_k^t - \log p_{i,k}^t \right.$$

$$\left. + \text{CrossEntropy}(p_i^t, g^t) - \text{Entropy}(p_i^t) \right]$$

Noting that

$$\text{CE}(p_i^t, g^t) - \text{Entropy}(p_i^t) = \text{KL}(p_i^t \,\|\, g^t)$$

the derivative can be written compactly as

$$\frac{\partial (L_{\text{adapt}}^t + L_{\text{reg}}^t)}{\partial z_{i,k}^t} = \frac{p_{i,k}^t}{N} \left[ \log \frac{g_k^t}{p_{i,k}^t} + \text{KL}(p_i^t \,\|\, g^t) \right]$$

## B  DETAILED HYPERPARAMETER SEARCH RESULTS

To complement the aggregate plots shown in the main text, Figs. 5-8 provide a per-direction break-down of the same search over the weighting coefficients $a_1$ and $a_2$ on the OFFICE-31 benchmark, together with an explicit visualization of the gap between the best-epoch and final-epoch accuracies.

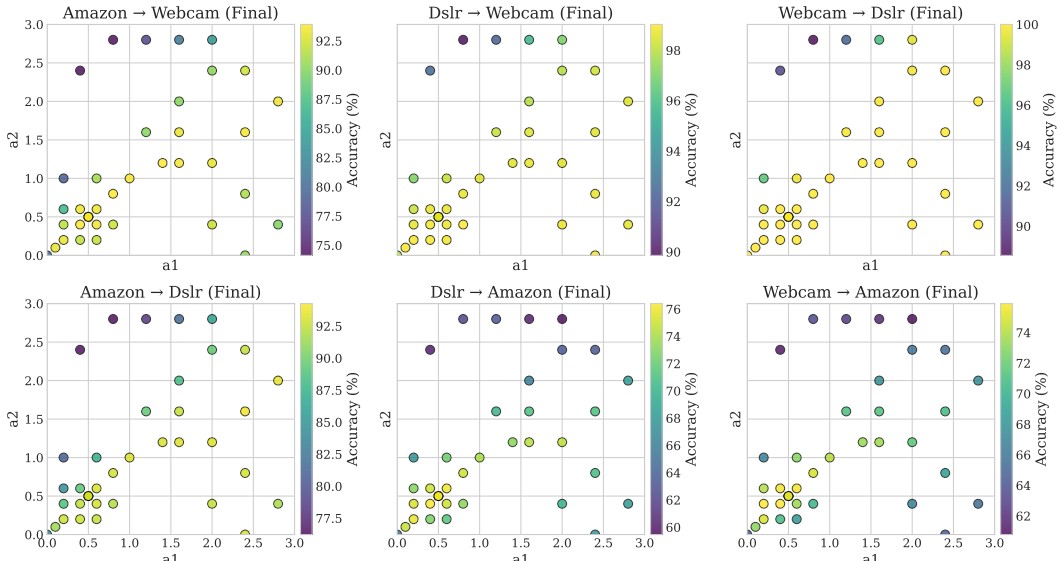

Figure 5: Final-epoch accuracy for the six Office-31 source→target tasks. Each dot shows the target-domain accuracy at the *last* epoch for one $(a_1, a_2)$ setting; colors fade from yellow (high) to purple (low). The best region aligns with the diagonal $a_1 = a_2$. Accuracy on the two hardest tasks (DSLR→Amazon and Webcam→Amazon) already drops once both weights exceed about $0.5$.

**Fig. 5. Final-epoch accuracy by adaptation task** For the difficult *DSLR→Amazon* and *Webcam→Amazon* adaptations the model's target accuracy is still modest midway through training. When both $a_1$ and $a_2$ rise above roughly $0.5$, the combined contribution of the unsupervised losses $L_{\text{adapt}}^t$ and $L_{\text{reg}}^t$ begins to dominate as the epoch-dependent scaling factor $\lambda$ grows. This dominance diminishes the supervised term $L_{\text{fid}}^s$, steering optimization toward over-confident yet inaccurate target predictions and neglecting the labeled source data. The net effect is the learning of spurious features and, ultimately, divergence on these harder source-target tasks. For *DSLR→Webcam* and *Webcam→DSLR*, target accuracy stays near $100\,\%$ across nearly all hyper-parameter combinations below (and even slightly above) the diagonal $a_1 = a_2$. Mid-training, the classifier is already highly accurate on the target data thanks to the strong influence of $L_{\text{fid}}^s$; the regularization loss $L_{\text{reg}}^t$ therefore contributes little, and $L_{\text{adapt}}^t$ alone suffices to align the domains. As a result, further diminishing the weight $a_2$ on $L_{\text{reg}}^t$ does not destabilize optimization, and accuracy remains consistently high throughout this region.

**Figure 6. Best-epoch accuracy by adaptation task** The highest target-domain accuracy is generally achieved near the diagonal $a_1 = a_2$, confirming that the two unsupervised losses perform best when their weights are balanced. Notable exceptions are the easy tasks *DSLR→Webcam* and *Webcam→DSLR*, where accuracy remains close to $100\,\%$ for almost all hyper-parameter combinations *below* the diagonal; as discussed earlier, once the classifier is already highly accurate, further reducing the relative influence of $L_{\text{reg}}^t$ does not destabilize training.

**Figure 7. Mean stability gap** The map shows how far accuracy falls from its peak by the end of training. When both $a_1$ and $a_2$ stay below about $0.8$, the supervised fidelity loss $L_{\text{fid}}^s$ still guides learning and the stability gap stays under three percentage points. Once the two coefficients rise past roughly $1.5$, the combined influence $a_1\lambda L_{\text{adapt}}^t + a_2\lambda L_{\text{reg}}^t$ overwhelms $L_{\text{fid}}^s$; late-epoch accuracy then drops by eight to twelve points, confirming the instability mechanism noted for Fig. 5.

**Figure 8. stability gap by adaptation task** The largest gaps appear on the two most difficult tasks, *DSLR→Amazon* and *Webcam→Amazon*. When both $a_1$ and $a_2$ rise above about $0.5$ along the $a_1 = a_2$ diagonal, the unsupervised losses $L_{\text{adapt}}^t$ and $L_{\text{reg}}^t$ increase with $\lambda$. Their growing influence diminishes the supervised term $L_{\text{fid}}^s$, pushes optimization toward overconfident yet inaccurate target predictions, and causes the model to neglect the labeled source data. The result is the learning

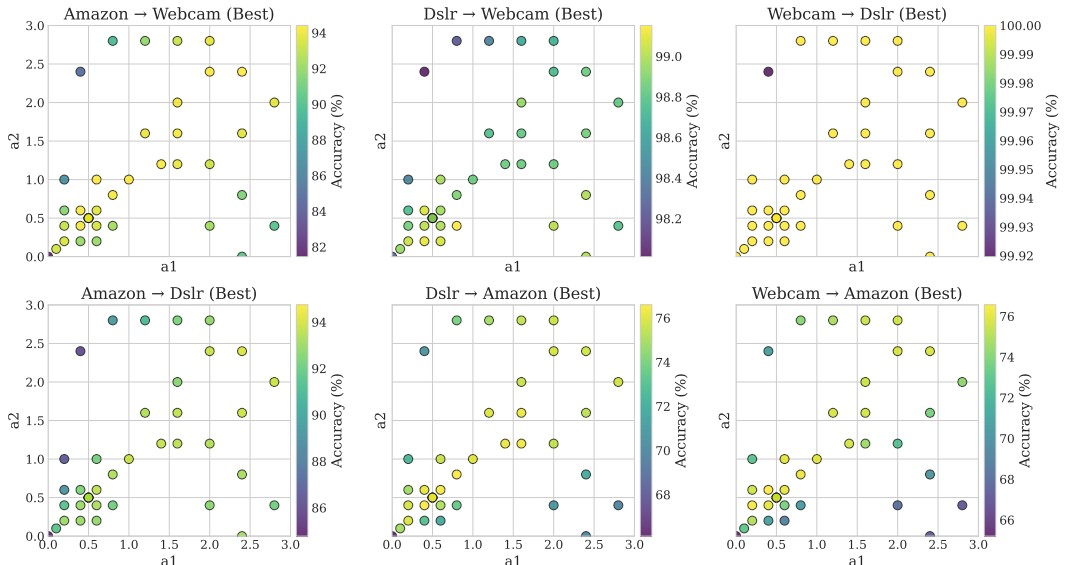

Figure 6: Best-epoch accuracy for the six Office-31 source→target tasks. Each dot marks the highest target-domain accuracy achieved at any epoch for a single $(a_1, a_2)$ combination; colors fade from yellow (high) to purple (low). For all six adaptation pairs the optimal region clusters along the diagonal $a_1 = a_2$, except that DSLR→Webcam and Webcam→DSLR remain high across much of the sub-diagonal region.

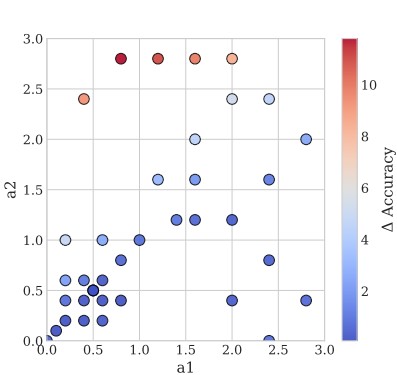

Figure 7: Mean stability gap for the six Office-31 source→target tasks. Each dot reports the gap $\Delta\mathrm{Acc} = \mathrm{Best} - \mathrm{Final}$ averaged over all six tasks for one $(a_1, a_2)$ setting; colors fade from blue (small drop) to red (large drop). The gap is smallest inside the pocket where $a_1$ and $a_2$ are both below about 0.8 and close to the diagonal $a_1 = a_2$; it grows quickly once the two coefficients exceed roughly 1.5, signaling late-stage instability when the unsupervised losses outweigh $L_{\mathrm{fid}}^s$.

of spurious features and, in these harder source-to-target tasks, the final-epoch drop becomes even larger. Other tasks avoid this issue because their target accuracy is already high, which supplies a more reliable unsupervised signal in the target-related loss terms.

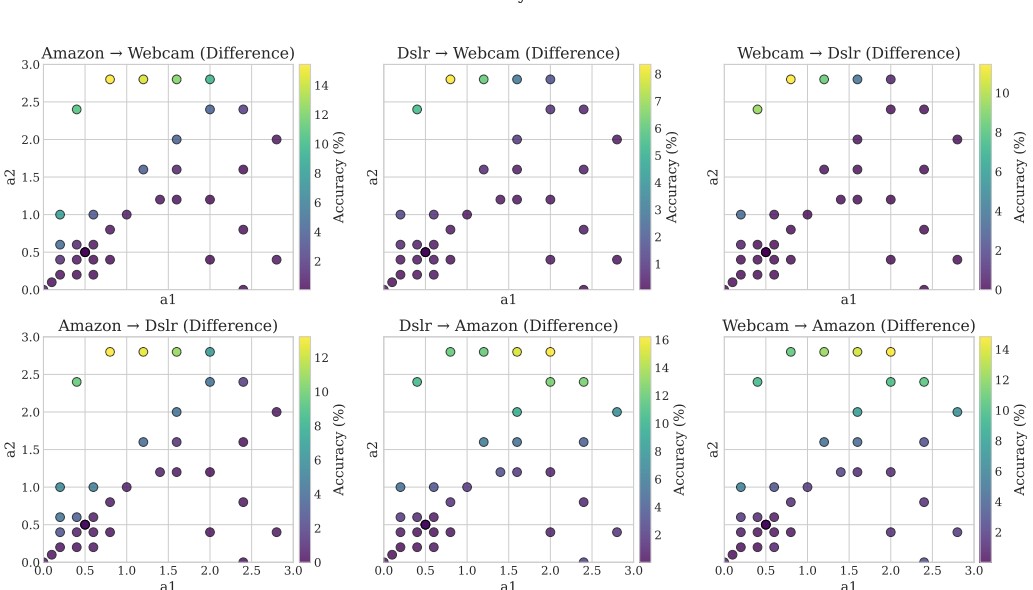

Figure 8: Stability gap for each Office-31 adaptation task. Each panel shows $\Delta\text{Acc} = \text{Best} - \text{Final}$ for one source→target pair. Each dot represents a single $(a_1, a_2)$ setting; colors fade from blue (small drop) to red (large drop).

## C EXPERIMENTAL SETTING

### C.1 DATASETS

Office-31 contains 4,110 images from 31 categories distributed across three distinct domains: Amazon (A), DSLR (D), and Webcam (W). Office-Home, a more complex dataset, comprises 15,588 images across 65 categories spanning four distinct domains: Art (Ar), Clipart (Cl), Product (Pr), and Real-World (Rw). We also evaluate on the VisDA-2017 benchmark, which is a large-scale domain adaptation dataset with 12 categories, where the source domain consists of synthetic renderings and the target domain consists of real-world images.

MNIST consists of 70,000 28×28 grayscale digit images, with 60,000 for training and 10,000 for testing, featuring clean, centered digits. USPS provides 9,298 16×16 grayscale digit images collected from U.S. postal mail, with 7,291 training and 2,007 test samples, and exhibits handwriting styles that differ from MNIST. SVHN contains color digit images from house numbers in natural scenes; the standard training set has 73,257 images and the standard test set has 26,032 images, with cluttered backgrounds and illumination changes that make recognition more challenging. For the digit datasets, we adopt the network architectures used by Ganin & Lempitsky (2015).

### C.2 SVHN→MNIST

For this adaptation, we use the standard training sets of both datasets and evaluate on the standard MNIST test set. The feature extractor comprises three convolutional layers (5×5 kernels, stride 1, padding 2), with 3×3 max-pooling (stride 2) after the first two layers, and concludes with a fully-connected block yielding 3072 features. The classifier is a two-layer MLP with 2048 hidden units and 10 output neurons. Batch normalization follows every layer, and dropout (rate = 0.5) is applied after the first fully-connected layer.

### C.3 MNIST→USPS & USPS→MNIST

Following Long et al. (2013), we sampled 2000 images from MNIST and 1800 from USPS for training, evaluating on each dataset's standard test split. For the MNIST*→USPS* experiment, we use the full standard training sets of both datasets. The feature generator includes two convolutional layers (5×5 kernels, stride 1) each followed by 2×2 max-pooling (stride 2), and a final fully-connected block projecting to 100 features. The classifier is again a two-layer MLP with 100 hidden units and 10 output neurons. As before, batch normalization is applied after every layer and dropout (rate = 0.5) after the first fully-connected layer.

For digit datasets, the second augmentation for $L_{ss}^t$ combines random affine transformations with random brightness and contrast perturbations.

For adaptation tasks on digit datasets (SVHN→MNIST, MNIST→USPS, and USPS→MNIST), target images were left unmodified for the base augmentation. For the second augmentation, we performed random affine transformations and random brightness and contrast adjustments to the target data used for the $L_{ss}^t$.

### C.4 OFFICE-31 & OFFICE-HOME

For these datasets, we use ResNet-50 as a feature extractor, followed by a two-layer MLP with 512 hidden neurons and an output layer sized to the number of classes in each dataset.

For these datasets, the base augmentation is a combination of random crop and random horizontal flip, which provides varied spatial transformations. The second augmentation introduces additional diversity through the inclusion of grayscale, random crop, and random horizontal flip.

### C.5 VISDA

For this dataset, we use ResNet-101 as a feature extractor, followed by a two-layer MLP with 512 hidden neurons and an output layer sized to the number of classes.

For this dataset, the base and second augmentations are the same as in Office-31.

### C.6 ADDITIONAL PREPROCESSING CONSIDERATIONS

For all experiments, input images were normalized by subtracting the per-channel mean and dividing by the per-channel standard deviation. When using ImageNet-pretrained models, we applied the standard ImageNet statistics. For models trained from scratch, we computed the per-channel mean and standard deviation on the source domain's training set and used those values to normalize inputs in both the source and target domains.

For these experiments, even with a fixed pseudo-random seed, we observe run-to-run performance variability due to algorithmic nondeterminism in cuDNN kernels, which employ atomic operations and parallel reduction orders that are not fixed. The cuDNN benchmark causes the library to profile multiple convolution implementations at runtime and select the fastest, but this selection can differ across runs and devices Contributors (2025). Additionally, variations in GPU micro-architecture, driver, and CUDA/cuDNN versions can alter low-level instruction scheduling and floating-point rounding behaviors, introducing further minor discrepancies Whitehead & Fit-Florea (2011). Together, these sources of nondeterminism explain why performance metrics may shift slightly despite using a fixed seed.

