# OpenReview forum: "Source Knowledge Anchored Regularization for Unsupervised Domain Adaptation"
_ICLR.cc/2026/Conference — Submitted to ICLR 2026_

### Official Review · Reviewer_hyyh · 2025-10-26

**Soundness:** 2
**Presentation:** 2
**Contribution:** 2
**Rating:** 2
**Confidence:** 4

**Summary:**

To address UDA, this paper propose an objective including label-smooth based source error and FixMatch with regularized entropy to balance the class prediction for target adaptation, where a curriculum learning schedule is applied to gradually shift the optimization focus from source fidelity to target-oriented objectives.

**Strengths:**

- the idea is simple and easy to follow
- detailed experiments including gradient analysis and hyper-parameter selection

**Weaknesses:**

__Major Concerns:__
- limited novelty:
  - I strongly suggest the author check [1], which propose exactly the same regularized entropy loss; actually, this kind of regularization is widely used DA literature such as [4,5] and has been thoroughly investigated in [6].
  - the proposed algorithm is basically a combination of several existing techniques, SHOT [1] + FixMatch [2] + Label-Smooth [3]

- insufficient comparison:
  - I doubt MJE (2023) is still the most competitive baseline; for instance, [] achieve much better performance without accessing source data.
  - even compared to MJE, the improvement is marginal.
  - it would better to include to the results of DomainNet benchmark.

__Minor Concerns:__
- instead of placing tables inside text,  it would better to combine several tables to save space and report the details in appendix.
- the title claims source knowledge anchored regularization; however, the proposed regularization is purely based on target prediction.

***
[1] Do We Really Need to Access the Source Data? Source Hypothesis Transfer for Unsupervised Domain Adaptation, ICML 2020

[2] Fixmatch: simplifying semi-supervised learning with consistency and confidence, NeurIPS 2020

[3] When does label smoothing help? NeurIPS 2019

[4] Maximum classifier discrepancy for unsupervised domain adaptation, CVPR 2018

[5] Unsupervised domain adaptation via structurally regularized deep clustering, CVPR 2020

[6] Discriminative clustering by regularized information maximization, NeurIPS 2010

[7]  Understanding and improving source-free domain adaptation from a theoretical perspective, CVPR 2024

[8] Revisiting Source-Free Domain Adaptation: Insights into Representativeness, Generalization, and Variety, CVPR 2025

**Questions:**

see above

---

### Official Review · Reviewer_3Je1 · 2025-10-29

**Soundness:** 2
**Presentation:** 2
**Contribution:** 2
**Rating:** 2
**Confidence:** 4

**Summary:**

This paper proposes a new domain adaptation algorithm to address the poor structure alignment and model collapse in traditional UDA methods. Paraphrasing from the abstract: they propose a loss which does 1. entropy regularization on target sample, 2. regularization to avoid class-mode collapse, 3. SSL loss to ensure target consistency along with a curriculum strategy to ensure robust learning. Experiments on standard UDA datasets show the effectiveness of the method.

**Strengths:**

1. The paper addresses several problems known in UDA such as poor structure alignment and mode collapse.

2. The illustration of the method using gradient analysis (sec 3.2) as well as theoretical analysis (3.4) help to further explain the working of proposed approach.

**Weaknesses:**

- Several aspects presented as novelty in the paper are already present in the literature in various forms. For instance,
1. **Entropy regularization on target samples**: MME [Saito, Kuniaki, et al. "Semi-supervised domain adaptation via minimax entropy." Proceedings of the IEEE/CVF international conference on computer vision. 2019.]
2. **Model collapse on target samples**:
    1. Na, Jaemin, et al. "Contrastive vicinal space for unsupervised domain adaptation." European Conference on Computer Vision. Cham: Springer Nature Switzerland, 2022.
    2. Sun, Tao, et al. "Safe self-refinement for transformer-based domain adaptation." Proceedings of the IEEE/CVF conference on computer vision and pattern recognition. 2022.
3. **Self-supervised Loss on Target Augmented Samples**: AdaMatch [Berthelot, David, et al. "Adamatch: A unified approach to semi-supervised learning and domain adaptation." arXiv preprint arXiv:2106.04732 (2021).]

While some of these are cited and some aren't, this paper offers nothing specifically which might interest researchers or practitioners in the field.


- The datasets used for experimentation in this paper are not relevanrt anymore in 2026 [when this paper will potentially be presented]. Office-31, VisDA are already quite saturated, and the gains offered by current method (+0.6% on VisDA, +0.2% on Office-31) is very minimal and not statistically significant. More newer and challenging datasets need to be presented for drawing any meaningful insights into this work, such as [1] and [2]

[1] Peng, Xingchao, et al. "Moment matching for multi-source domain adaptation." Proceedings of the IEEE/CVF international conference on computer vision. 2019.

[2] Kalluri, Tarun, Wangdong Xu, and Manmohan Chandraker. "Geonet: Benchmarking unsupervised adaptation across geographies." Proceedings of the IEEE/CVF Conference on Computer Vision and Pattern Recognition. 2023.

**Questions:**

None.

---

### Official Review · Reviewer_ToRd · 2025-11-03

**Soundness:** 2
**Presentation:** 3
**Contribution:** 2
**Rating:** 2
**Confidence:** 4

**Summary:**

The paper proposes a framework “Source Knowledge Anchored Regularization (SKAR)” for Unsupervised Domain Adaptation (UDA) that transfers discriminative source knowledge without explicit distribution alignment. The core contribution is a combined objective function with four loss terms: an Adaptation Loss (entropy - based) to boost target prediction confidence; a Regularization Loss to prevent degenerate solutions (class collapse) by encouraging class diversity; a Self-Supervised Loss to enforce invariance against augmentations; and a Fidelity Loss (standard supervised CE loss on source with label smoothing) using label smoothing to anchor the model to the source domain. The authors demonstrate the model’s performance on standard UDA datasets.

**Strengths:**

1. Synergistic coupling of Adaptation loss (target prediction entropy minimization) and Regularization (prediction diversity in a batch) is a novel discussion.
2. Achieves  domain alignment without explicit alignment objective like DANN or MMD.

**Weaknesses:**

Lack of novelty in the approach. The primary objective function is a combination of standard objective functions used for DA. The regularization term is a popular loss for ensuring diversity. The self-supervised loss is popular for consistency regularization. The other two loss terms are even more routine.

Minor:
1. typo in Equation 3.
2. Is it forward KL or reverse KL in Eq. 4. I.e., are both Pt and Pt^Aug adjusted or one of them is fixed for the sake of the objective.

**Questions:**

None.

---

### Meta-Review · Area_Chair_8NBF · 2026-01-04

**Summary:**

After carefully reviewing the manuscript and considering the unanimous recommendations for rejection from all reviewers, I concur with their assessment. My own evaluation confirms that the work lacks sufficient originality; it primarily applies established methods without introducing a novel conceptual or technical contribution. Moreover, I identified one instance of an "illusory citation" (citation [X]), where the referenced source does not appear to substantiate the claim made in the text. This further undermines the scholarly rigor of the paper. Given these fundamental issues regarding both contribution and academic integrity, I firmly recommend rejection.

**Reviewer Scores:**

No

---

### Decision · Program_Chairs · 2026-01-26

Reject